# Pricing and coordination in a dual-channel supply chain with a socially responsible manufacturer

**Shiyang Li[1], Mengli Li** **[1]\*, Nan Zhou[2,3]**

**1** School of Economics and Management, Chongqing University of Posts and Telecommunications, Chongqing, PR China, **2** School of Chemistry and Chemistry Engineering, Yangtze Normal University, Chongqing, PR China, **3** Research Center for Economy of Upper Reaches of the Yangtze River, Chongqing Technology and Business University, Chongqing, PR China

\* limengli93@163.com

**Data Availability Statement:** All relevant data are within the paper and its Supporting Information File.

**Funding:** This research is supported by Ministry of Education, Humanity, and Social Science Research

## Abstract

This paper aims at designing coordination contract in a dual channel supply chain (DCSC) which consists of a socially responsible manufacturer and a retailer. We build stylized game models under both centralized and decentralized scenarios. Then, we identify the reason for supply chain inefficiency under decentralized scenario. Further, according to the manufacturer's corporate social responsibility (CSR) coefficient, we design two different contracts to achieve coordination. We find that with the impact of CSR, social welfare under centralized scenario is always higher than that under decentralized scenario. However, profit of the whole supply chain between the two scenarios has different relationship. More specifically, when CSR coefficient is relatively low, profit under centralized scenario is higher than that under decentralized scenario. When CSR coefficient is high, profit under centralized scenario is lower than that under decentralized scenario. Due to these two cases, we respectively design revenue sharing contract with franchise fee and wholesale price contract with franchise fee and government subsidy to achieve coordination. The result suggests that encouraging the manufacturer to bear CSR properly can reach a multi-win for social welfare, consumers and supply chain members through coordination contract. However, when CSR coefficient is higher than a certain threshold, conflict between supply chain members becomes irreconcilable which results in the retailer's resistance. In this condition, only through subsidy from government or philanthropic organization can supply chain members sustain their cooperation.

## 1. Introduction

Currently, sustainable development has received wide attentions from entrepreneurs and scholars [1, 2]. And increasing firms believe that bearing corporate social responsibility (CSR) is a significant way to achieve sustainable development [3]. According to a survey, 90% of firms among Fortune 500 around the world regard CSR as a core component of their goals [4].

Project [grant number 19YJC630084], Scientific and Technological Research Project of Chongqing Municipal Education Committee [grant number KJQN201800627], and Social Science Planned Project of Chongqing Municipal [grant number 2018QNGL35]. The first two funds is obtained by first author; the third fund is obtained by third author.

**Competing interests:** No authors have competing interests.

On the one hand, some firms are widely condemned due to the lack of CSR [5]. For example, Foxconn Technology Group was accused of highly intensive working conditions and militarized management, which caused several young employees to jump off a building. On the other hand, more and more consumers would like to purchase the products produced by socially responsible manufacturers. It is shown that 70% of consumers would like to pay an extra price for these products, which leads to more profit of these manufacturers [6]. Therefore, more and more manufacturers take part in designing socially responsible business operation [7, 8]. For example, in China, Volkswagen launched "Child Safety Action" to promote consumers to use child safety seats; HP has become the first batch enterprises to implement an environmental management system (EMS) to save energy and recycle used products, and has become one of the international enterprises certified by ISO 14001.

Meanwhile, with the growth of e-commerce, thousands of manufacturers begin to sell products online. At the same time, they sell through the traditional distribution channel in the highly competitive global market [9]. Generally speaking, traditional channel and online direct channel have different competitive edges. Thus, firms adopting the dual-channel strategy often occupy higher market share than those singly adopting the traditional channel or the online channel. A survey shows that nearly 42% of the famous manufacturers, including HP, IBM, and Apple, sell products to consumers directly through e-commerce [10]. However, the dual-channel strategy results in channel competition and conflict, which leads to the dilution effect between channels. For example, in December 2015, traditional pharmacies in India organize a strike of 800,000 pharmacists to ask the government to crack down the online retails of medicine. If the manufacturer has CSR, the openness of online direct channel breaks the balance between economic performance and socially responsible performance, which brings higher uncertainty for the supply chain's sustainable development. Specifically, the dual-channel strategy brings horizontal competition, which results in the resistance of traditional retailers. Will this conflict aggravate or alleviate when the manufacturer has CSR? Besides, having CSR can cause short term economic loss, which results in a higher contradiction between supply chain members. Will the socially responsible manufacturer that adopts the dual-channel strategy have the incentive to alleviate or eliminate this contradiction?

Coordination contract is efficient to mitigate or eliminate the conflicts among supply chain members. There has extensive literature that studies coordination contract design in a dual channel supply chain (DCSC for short) [11–17]. However, these researches mainly design coordination contract where all supply chain members aim at maximizing profit. To the authors' best knowledge, little literature has explored how to design coordination contract in a DCSC when the manufacturer has CSR concern. But in reality, increasing manufacturers that adopt dual-channel strategy have become more socially responsible. For example, L'Oreal, one of the world's largest cosmetics companies, cooperates with Alibaba Group to use eco-friendly packaging for all products sold online in China. Thus, studying coordination contract design in a DCSC with a socially responsible manufacturer has important implications for these firms, which is also the main contribution of our paper.

Under this background, we consider that a manufacturer with CSR concern adopts the dual-channel strategy to sell product. Then we build a game-theoretical model in two scenarios (centralize vs decentralized). After that we get equilibrium decisions of both the socially responsible manufacturer and the retailer. And then, we discuss the influence of CSR coefficient of the manufacturer. Through comparing equilibrium results between the two scenarios, we explore the reason why the supply chain suffers inefficiency when the manufacturer has CSR under decentralized scenario. Then to maximize social welfare, we propose internal coordination and external compensation coordination to achieve coordination on the basis of CSR coefficient.

The rest of this paper is arranged as following. Section 2 is a literature review. Section 3 is a basic model description. Then, section 4 compares equilibrium results between centralized and decentralized scenarios to explore the reason for supply chain inefficiency under decentralized scenario. Section 5 proposes coordination contracts when the manufacturer has low and high CSR coefficient, respectively. Section 6 gives conclusion and significance of this paper.

## 2. Literature review

Our paper relates to two distinct areas of the existing literature: CSR; coordination contract design in a DCSC.

### 2.1 CSR

With wide attention to the environment and sustainable development, CSR becomes a hot topic in academic circles. Some scholars mainly explore the effect of CSR on firms' financial income, but the academic research conclusions have been inconclusive. For instance, Lins et al. [18] point out that firms with high CSR behaves better on profit, growth, and sales than those with low CSR, i.e., CSR can promote ss performance. However, some scholars believe that there is negative correlation [19] or nonlinear correlation [20] between CSR and firms' performance. There are two differences between this paper and the above researches. First, the above researches are mainly qualitative or empirical, but we build a game model to examine the relationship between CSR and corporate decision-making. Second, the above researches do not put firms in the supply chain environment. However, in practice operational decisions of firms are often influenced by partners within the same supply chain.

Under the supply chain environment, CSR is especially important to meet business needs [21]. That is because that CSR affects not only the purchase intention of consumer [7] but also the social reputations of firms [22, 23]. Therefore, CSR has emerged frequently under supply chain environment. However, these studies mainly maximize individual benefits of supply chain members, but fail to study how to use coordinating contract to maximize overall supply chain benefits. In recent years, a small body of literature designs contracts to coordinate a supply chain with socially responsible firms. For instance, Ni et al. [24] establish mathematical models to design coordination contracts when a supplier has CSR and when a retailer has CSR, and find that economic performance and CSR performance have inherent conflict and cannot achieve the maximization of both; Hsueh [25] considers a manufacturer has CSR, and designs a revenue sharing contract to achieve maximization of profit and CSR performance. The above literature regards a specific behavior such as environmental protection and charity activities as a firm's CSR behavior and discusses the decision on CSR behavior and coordination.

The most relevant literature on CSR to this paper uses consumer surplus to simulate the impact of CSR when firms make the operation decisions. Panda [26] considers two scenarios where the manufacturer has CSR and where the retailer has CSR, and respectively proposes revenue sharing contracts to maximize profit of the supply chain. Panda et al. [6] study a three-layer supply chain with a socially responsible manufacturer, and propose a contract to resolve channel conflict and achieve coordination of this supply chain. In view of the idea in the above literature, this paper also incorporates consumer surplus into the socially responsible manufacturer's optimal decisions. But our paper differs the above researches from two aspects. First, the previous researches mainly focus on coordination contract design in a single channel supply chain and do not study the coordination problem in a DCSC when the manufacturer has CSR. However, there exists not only vertical competition but also horizontal channel

competition in a DCSC, which makes it more complicated to study coordination contract. Second, designing a coordination contract in previous literature is based on profit transfer within a supply chain. However, our paper finds that when a manufacturer has high CSR coefficient, profit transfer within a supply chain system fails to coordinate. To solve this problem, we propose a wholesale price contract with franchise fee and government subsidy based on the practice that the Chinese government offers a subsidy to the firms that undertake CSR.

## 2.2. Coordination contract design in a DCSC

As e-commerce develops rapidly in recent years, increasing manufacturers implement dual-channel strategies to sell products. Therefore, increasing scholars study operational decisions in a DCSC [27, 28]. They found that dual-channel strategy breaks the traditional supply chain's balance and results in conflict in two channels. In a traditional supply chain, large scholars pay attention on coordination contract design to solve the problem caused by individual rationality and double marginalization [29]. Thus, some scholars use internal contract proposed in a traditional supply chain to coordinate conflict in a DCSC. For instance, some scholars use cooperative advertising [16, 30], inventory cooperation [31], and service cooperation [13] to coordinate the DCSC.

Most of the literature studies coordination contract design in a DCSC through pricing. For instance, Chen et al. [12] study equilibrium pricing decisions of supply chain members in a DCSC, and design a joint pricing contract to achieve coordination; Cao [11] considers the risk of demand disruption in a DCSC, proposes a revenue sharing contract and point out that the effectiveness of this contract is influenced by the joint impact of several key factors. The above literature mainly focuses on risk-neutral supply chain members, but in practice some medium-sized and small firms cannot bear strong risk. Thus some scholars consider the impact of risk-averse. For instance, considering the retailer is risk-averse in a dual-channel supply chain, Li et al. [14] study risk sharing contract design when market demand faces uncertainty; Xu et al. [17] consider a DCSC which consists of a risk-averse supplier and a risk-averse retailer, and design a revenue sharing contract to get a win-win situation. Besides, some scholars introduce competition between traditional retailers in a DCSC. David and Adida [32] consider a supply chain consisting of several traditional retailers and a supplier that operates an online channel, and find that quantity contract fails to achieve coordination but can achieve performance improvement. Furthermore, as sustainable development becomes a hot trend, increasing scholars pay attention to coordination contract design in a DCSC with closed-loop situation or green concern. For instance, Zheng et al. [33] consider different channel power structures in a DCSC with closed-loop situation and two-part tariff contracts had been designed to obtain the maximum profit of the whole supply chain; Aslani and Heydari [34] study pricing decision and product greenness decision in a DCSC which faces channel disruption, and design a contract to eliminate conflicts of supply chain members; Ranjan, and Jha [35] consider a manufacturer sells an eco-friendly product through online channel and a traditional product through offline channel, study revenue-sharing mechanism in this dual-channel supply chain.

The above literature on coordination contract design in a DCSC through pricing is more related to this paper. But the manufacturers in the above researches mainly aim at profit maximization, and they mainly study coordination contract design according to the largest profit obtained from centralized scenario. Differently, we consider that the manufacturer undertakes CSR, which is a key factor for the manufacturer to achieve sustainable development. Besides, due to undertaking CSR, we study coordination contract design in a DCSC according to social welfare maximization.

## 3. Problem description

Consider a DCSC which consists of a socially responsible manufacturer ($m$) and a retailer ($r$). The manufacturer sells the product to the retailer at wholesale price $w$. And then, the product is sold by the retailer to the consumer at selling price $p_r$. Meanwhile, the manufacturer operates its online store, and sells the same product to the consumer directly at direct price $p_d$. Let $c_r$ and $c_d$ denote constant marginal selling cost of the retailer and the online store of the manufacturer. For analytical simplicity, similar to [36], we normalize the market size of the product to 1 and the manufacturer's constant production cost to 0.

When buying the product from the retailer, the consumer's willingness to pay is $v$. Considering consumer heterogeneity on willingness to pay, this paper supposes that $v$ is uniformly distributed between 0 and 1. Then the net consumer utility of purchasing the product from the traditional channel as $U_r = v-p_r$. When buying the product from the manufacturer's online store, the consumer may suffers various disadvantages, including the discrepancy between virtual and physical inspections of the product, hassle cost caused by product return or product exchange and delay risk of product delivery. Therefore, we suppose the willingness of consumer when buying from the online store is $\theta v$, where $\theta$ denotes acceptance of the manufacture's online store of the consumer and satisfies $\theta \in (0,1)$. Then we have the net consumer utility of purchasing the product from the online store as $U_d = \theta v - p_d$, which is a widely accepted assumption in the literature [37]. According to the utility maximization principle, the consumer makes decisions about whether and where to buy. When $v \in \{v|U_r \geq 0, U_r \geq U_d\}$, i.e. $(p_r-p_d)/(1-\theta) \leq v < 1$, the consumer purchases the product from the traditional channel; when $v \in \{v|U_d \geq 0, U_d \geq U_r\}$, i.e. $p_d/\theta \leq v < (p_r-p_d)/(1-\theta)$, the consumer purchases the product from the manufacture's online store; when $v \in \{v|U_r < 0, U_d < 0\}$, i.e. $0 < v < p_d/\theta$, the consumer leaves the market without purchasing anything. The above purchasing behavior of the consumer is also given in Fig 1.

Based on this, it is easy to get consumers' demand of the traditional channel as $q_r = 1 - (p_r-p_d)/(1-\theta)$ and consumers' demand of the manufacturer's online store as $q_d = (p_r-p_d)/(1-\theta)-p_d/\theta$. To discourage the retailer purchasing the product from the manufacture's online store, this paper supposes $p_d > w$. Besides, to make sure demand of direct channel is positive, we assume $q_d > 0$, i.e. $\theta p_r > p_d$. Then profits of the manufacturer and the retailer are,

$$\pi_m = wq_r + (p_d - c_d)q_d \tag{1}$$

$$\pi_r = (p_r - w - c_r)q_r \tag{2}$$

Recently, Company Law and Labor Contract Law enacted by the Chinese Government make it clear that firms need to fulfill social responsibility. Many firms face intense pressure on how to manage socially responsible operations. As is known to all, manufacturers are always

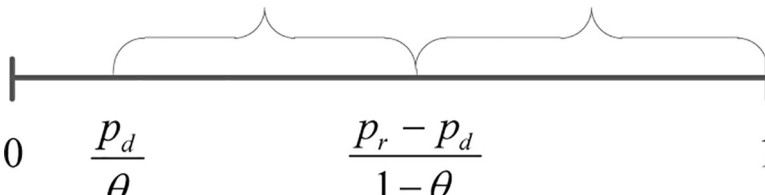

**Fig 1. The consumer purchasing behavior under the dual-channel strategy.**

the core company in most supply chains, so the main pressure brought by CSR in the supply chain lies with the manufacturers. Thus, we suppose that the manufacturer exhibits CSR in this article, i.e. the manufacturer makes decisions based on social welfare. Social welfare is defined as the sum of the firm's profit brought by selling products and consumer surplus [6, 26]. Thus, we have optimization problem of the manufacturer is,

$$V_m = \pi_m + \alpha CS = wq_r + (p_d - c_d)q_d + \alpha CS \tag{3}$$

Where $CS = \int_{(p_r-p_d)/(1-\theta)}^{1} (v - p_r)dv + \int_{p_d/\theta}^{(p_r-p_d)/(1-\theta)} (\theta v - p_d)dv$; $\alpha$ denotes CSR coefficient of the manufacturer which satisfies $\alpha \in [0,1]$, the larger the CSR coefficient $\alpha$ is, the higher social responsibility of the manufacturer is, $\alpha = 0$ denotes the manufacturer is a complete profit maximization enterprise, $\alpha = 1$ denotes the manufacturer is a completely social responsibility maximization enterprise.

## 4. Discussion

In this section, we first obtain equilibrium decisions under centralized scenario (C scenario for short) and decentralized scenario (D scenario for short). Then, this paper explores the impact of CSR coefficient of the manufacturer on equilibrium decisions. Through the comparison of equilibrium results between the two scenarios, we explore the reason for supply chain inefficiency to provide theoretical support for supply chain coordination.

### 4.1. The centralized scenario (C)

Under C scenario, the supply chain makes decisions to optimize system-wide profit. The wholesale price is not a decision variable. The decision variables are only the selling prices of the product. Then according to (2) and (3), the optimization problem under the centralized scenario is:

$$\begin{cases} \max_{p_r, p_d} V^C = (p_r - c_r)q_r + (p_d - c_d)q_d + \alpha CS \\ p_d < \theta p_r \end{cases} \tag{4}$$

Solving the optimization problem in (4), we have theorem 1. The proof of theorem 1 and the other proofs of theorems and propositions are given in the appendix.

**Theorem 1.** The equilibrium pricing decisions of the DCSC under C scenario are:
$p_r^{C*} = \frac{1-\alpha+c_r}{2-\alpha}$ and $p_d^{C*} = \frac{\theta-\alpha\theta+c_d}{2-\alpha}$;
The equilibrium profit and consumer surplus of the DCSC are:
$\pi_{SC}^{C*} = \frac{(1-\alpha)\{\theta[(1-c_r)^2-2c_dc_r]-\theta^2(1-2c_r)+c_d^2\}}{\theta(1-\theta)(2-\alpha)^2}$, $CS^{C*} = \frac{\theta[(1-c_r)^2-2c_rc_d]-\theta^2(1-2c_r)+c_d^2}{2\theta(1-\theta)(2-\alpha)^2}$;
The social welfare of the supply chain is: $SW^{C*} = \pi_{SC}^{C*} + CS^{C*}$.

### 4.2. The decentralized scenario (D)

Under D scenario, supply chain members make decision to maximize their own profit. The socially responsible manufacturer decides both $w$ and $p_d$ firstly, following that the retailer decides $p_r$. Then according to (2) and (3), the optimization problems of supply chain members under D scenario can be expressed as:

$$\begin{cases} \max_{w, p_d} V_m^D = wq_r + (p_d - c_d)q_d + \alpha CS \\ \text{s.t. } p_d > w \\ \qquad p_d < \theta p_r \end{cases} \tag{5}$$

$$\max_{p_r} \pi_r^D = (p_r - w - c_r)q_r \tag{6}$$

For convenience, we define $C_1 = (1-\theta-c_r)(2-\alpha)/2$, $C_2 = \theta(2+2c_r-\alpha c_r-2\theta)/(4-2\theta-\alpha)$, and $C_3 = (1-\theta)(2-2\theta-\alpha)/(2-\alpha)$.

Using backward induction to solve formula (5) and (6), we have the following theorem 2.

**Theorem 2.** When $C_1 < c_d < C_2$ and $c_r > C_3$, the equilibrium pricing decisions are:

$w^{D*} = \frac{(2-\alpha)^2(1-c_r)-\alpha(\theta-c_d)}{(2-\alpha)(4-\alpha)}$, $p_d^{D*} = \frac{\theta(1-\alpha)+c_d}{2-\alpha}$, and $p_r^{D*} = \frac{(2-\alpha)(3-\alpha+c_r)-2(\theta-c_d)}{(2-\alpha)(4-\alpha)}$;

The manufacturer's equilibrium profit and total utility are:

$$\pi_m^{D*} = \frac{\left\{ \begin{array}{l} \theta(2-\alpha)^3[(1-c_r)^2 + 2c_r(\theta-c_d) - \theta] + \theta(\theta-2c_d)(1-\theta)(3\alpha^2 - 12\alpha + 8) \\ +\alpha^2 c_d^2(3-\alpha) + c_d^2(2-\theta)(3\alpha^2 - 12\alpha + 8) \end{array} \right\}}{\theta(1-\theta)(2-\alpha)^2(4-\alpha)^2}$$

and $V_m^{D*} = \frac{\theta c_r(c_r + 2\theta - 2 - 2c_d)(2-\alpha) + (1-\theta)[2(\theta-c_d)^2 + \theta(2-\alpha)] + c_d^2(2-\alpha)}{2\theta(1-\theta)(2-\alpha)(4-\alpha)}$.

The retailer's profit and consumer surplus are: $\pi_r^{D*} = \frac{(1-\theta-c_r+c_d)^2}{(1-\theta)(4-\alpha)^2}$;

$CS^{D*} = \frac{\theta(1-\theta)[4(\theta-2c_d)(3-\alpha)+(2-\alpha)^2] + \theta c_r(2-\alpha)^2(c_r+2\theta-2-2c_d) + c_d^2[(4-\alpha)^2 + 4\theta(\alpha-3)]}{2\theta(1-\theta)(2-\alpha)^2(4-\alpha)^2}$.

The supply chain's profit and social welfare are: $\pi_{SC}^{D*} = \pi_m^{D*} + \pi_r^{D*}$ and $SW^{D*} = \pi_m^{D*} + \pi_r^{D*} + CS^{D*}$.

Then we analyze the impact of CSR coefficient of the manufacturer $\alpha$, which is given in proposition 1.

**Proposition 1.** The impact of CSR coefficient $\alpha$:

1. $\frac{\partial w^{D*}}{\partial \alpha} < 0, \frac{\partial p_r^{D*}}{\partial \alpha} < 0, \frac{\partial p_d^{D*}}{\partial \alpha} < 0$;

2. $\frac{\partial q_r^{D*}}{\partial \alpha} > 0, \frac{\partial q_d^{D*}}{\partial \alpha} > 0$;

3. $\frac{\partial \pi_m^{D*}}{\partial \alpha} < 0, \frac{\partial CS^{D*}}{\partial \alpha} > 0, \frac{\partial V_m^{D*}}{\partial \alpha} > 0, \frac{\partial \pi_r^{D*}}{\partial \alpha} > 0$;

4. $\frac{\partial \pi_{SC}^{D*}}{\partial \alpha} < 0, \frac{\partial SW^{D*}}{\partial \alpha} > 0$.

Proposition 1 (1) shows that selling prices of both channels and wholesale price decrease as CSR coefficient of the manufacturer $\alpha$ increases. The intuition behind this phenomenon is as follows. At higher values of $\alpha$, the manufacturer takes into account consumer surplus more. As a result, the manufacturer cuts its wholesale price and selling price in online store. As a competitor, the retailer cuts its retail price too. Meanwhile, proposition 1 (2) shows that as the prices go down in both the traditional and direct channels, sales go up. It indicates that it is more conducive for product promotion when the manufacturer tends to be more socially responsible.

Proposition 1 (3) indicates that as the manufacturer has a higher CSR coefficient, its profit decreases but consumer surplus increases, which leads to an increase in its total utility. In practice, some manufacturers develop low-carbon or green manufacturing processes to fulfill social responsibility. Although these manufacturers suffer profit loss due to social responsibility, consumer surplus increases. Thus consumers are more likely to recognize these socially responsible manufacturers. Meanwhile, with higher CSR coefficient of the manufacturer, retail price decreases while selling quantity increases, which leads to a profit increase of the retailer. It indicates that the retailer is inclined to cooperate with the manufacturer that possesses higher social responsibility.

Proposition 1 (4) indicates that along with CSR coefficient of the manufacturer increases, the decrease of the manufacturer's profit dominates the increase of the retailer's profit, resulting in profit decreases of the supply chain. Besides, at higher values of CSR coefficient of the manufacturer, consumer surplus increases, which is exceeding the profit loss of supply chain. As a result, the whole social welfare increases. It indicates that under D scenario, the manufacturer fulfills more social responsibility, social welfare increases while its profit decreases.

### 4.3. Comparison between different scenarios

By comparing equilibrium results between D and C scenarios, we investigate the reason for the DCSC inefficiency under D scenario. Then we obtain proposition 2.

**Proposition 2.** The comparison of equilibrium results between D and C scenarios:

1. $p_d^{C*} = p_d^{D*}$ and $p_r^{C*} < p_r^{D*}$;

2. when $0 \leq \alpha < 2 - \sqrt{2}$, $\pi_{SC}^{C*} > \pi_{SC}^{D*}$; when $2 - \sqrt{2} \leq \alpha \leq 1$, $\pi_{SC}^{C*} < \pi_{SC}^{D*}$;

3. $SW^{C*} > SW^{D*}$.

Proposition 2 implies that selling price in online store under D scenario equals that under C scenario while retail price under D scenario is higher than retail price under C scenario. Proposition 2 also indicates that social welfare under D scenario is always lower than C scenario. The reason is as follows. Under C scenario, the social responsibility supply chain is an integrated entity, which makes decisions based on total utility maximization instead of profit maximization. Thus the social responsibility supply chain cuts the retail price to obtain optimal social welfare. Under D scenario, the socially responsible manufacturer makes decisions based on total utility maximization while the retailer is based on profit maximization. Due to the inconsistency of decision goals and double marginalization under D scenario, the DCSC suffers social welfare loss.

It is common sense that profit obtained under C scenario is always higher than that under D scenario in traditional supply chain. However, in the DCSC with a socially responsible manufacturer, when $\alpha < 2 - \sqrt{2}$, profit of the supply chain profit under C scenario is higher than that under D scenario; when $2 - \sqrt{2} < \alpha < 1$, profit of the supply chain profit under C scenario is lower than that under D scenario. The reason behind this phenomenon is as follows. With the increase of CSR coefficient of the manufacturer, profits of the supply chain decrease under both D and C scenarios. However, the descent speed of profit under C scenario is faster than that under D scenario.

To explain the effect of the manufacturer's CSR on the supply chain profits and social welfare under the two scenarios, we draw Figs 2 and 3 based on $\theta = 0.7$, $c_r = 0.3$, and $c_d = 0.2$.

## 5. Coordination contract design of DCSC

Under C scenario, the whole supply chain obtains the optimal social welfare. However, under D scenario, the whole supply chain suffers social welfare loss due to individual rationality. As the leader of the supply chain, the socially responsible manufacturer has incentive to design a contract to coordinate the supply chain. Furthermore, with the influence of socially responsibility, the purpose of the manufacturer to design contract are twofold: (1) to achieve the optimal social welfare under C scenario; (2) to make sure profits of supply chain members are no less than those under D scenario. Therefore, how to design coordination contract to achieve the above two purposes is the main problem facing by the socially responsible manufacturer. To be more specific, when $0 < \alpha < 2 - \sqrt{2}$, profit and social welfare of the DCSC under C

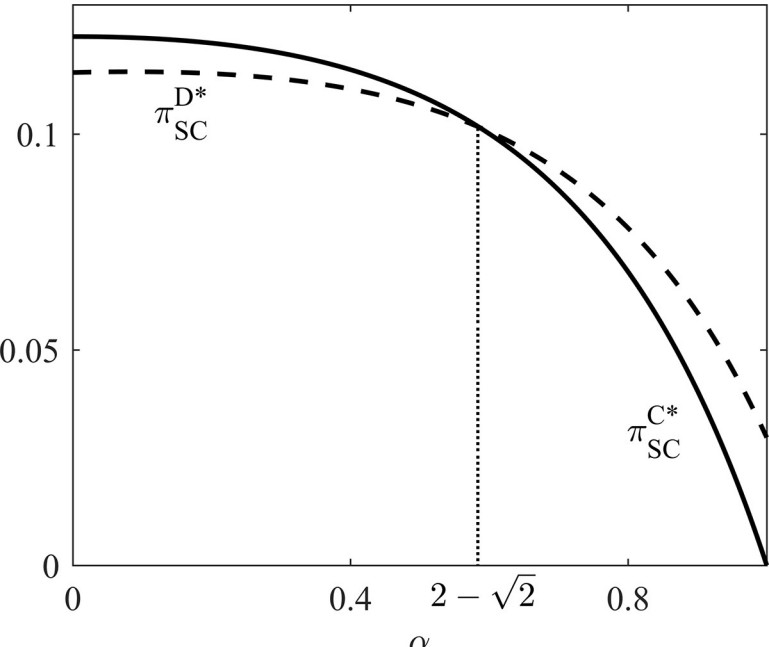

**Fig 2. Profit comparison between D and C scenarios.**

scenario are both greater than those under D scenario. Reasonable coordination mechanism achieves not only higher profits but also higher social welfare. When $2 - \sqrt{2} \leq \alpha \leq 1$, profit of the DCSC is lower while social welfare is higher than those under D scenario. It indicates that when CSR coefficient of the manufacturer is sufficiently high, internal transfer contract

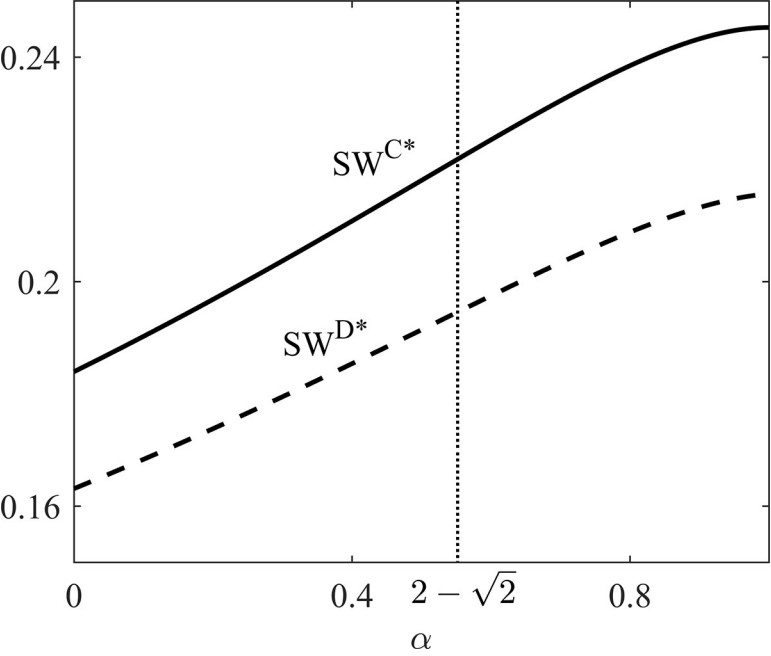

**Fig 3. Social welfare comparison between D and C scenarios.**

within a DCSC cannot satisfy supply chain members' participation constraints. To satisfy these participation constraints, the DCSC needs to receive an external subsidy. Thus we propose two contracts to coordinate the two situations mentioned above. For convenience, we refer to conditions $0 < \alpha < 2 - \sqrt{2}$ and $2 - \sqrt{2} \leq \alpha \leq 1$ as low (L) and high (H) CSR scenarios, respectively.

## 5.1. Revenue sharing contract with franchise fee under low CSR scenario (L)

As a leader of the DCSC, the socially responsible manufacturer offers a revenue sharing contract with franchise fee to the retailer to alleviate channel conflict. At the beginning of the selling season, the socially responsible manufacturer charges the retailer a franchise fee $F$ and then commits to a low wholesale price. After the selling season, the retailer gives the socially responsible manufacturer a certain proportion of its revenue, let $\lambda$ denote this revenue sharing rate.

In the revenue sharing contract with franchise fee, optimization problem of the manufacturer is:

$$\max_{p_d, w} V_m^L = wq_r + (p_d - c_d)q_d + \lambda p_r q_r + \alpha CS + F$$
$$\text{s.t. } p_d > w \tag{7}$$
$$\theta p_r > p_d$$

Where,

$$p_r^*(w, p_d) = \arg \max_{p_r} \pi_r^L = [(1-\lambda)p_r - w - c_r]q_r - F \tag{8}$$

Solving the optimization problem in (7) and (8), we obtain:

**Theorem 3.** In the revenue sharing contract with franchise fee, equilibrium results of the supply chain are: $w^{L*} = \frac{(\alpha-2\lambda)c_r + (1-\lambda)(\theta-\alpha-c_d)}{2-\alpha}$, $p_r^{L*} = p_r^{C*}$, $p_d^{L*} = p_d^{C*}$;

$\pi_m^{L*} = \frac{\theta\lambda(1-\theta+c_d-c_r)^2 + (1-\theta)(\theta-c_d)^2 - \alpha\{\theta^2(2c_r-1) + \theta[(1-c_r)^2 - 2c_r c_d] + c_d^2\}}{\theta(1-\theta)(2-\alpha)^2} + F^{L*}$,

$\pi_r^{L*} = \frac{(1-\lambda)(1-\theta+c_d-c_r)^2}{(1-\theta)(2-\alpha)^2} - F^{L*}$.

Theorem 3 indicates that when $0 < \alpha < 2 - \sqrt{2}$, the socially responsible manufacturer can use revenue sharing contract with franchise fee to achieve coordination. From $\partial w^{L*}/\partial\lambda < 0$, we see that when wholesale price is high, the socially responsible manufacturer will lower the revenue sharing rate to obtain optimal benefit of the supply chain.

On the basis of equilibrium decisions mentioned before, when revenue sharing contract with franchise fee is accepted, the DCSC can obtain the same profit and social welfare as those under C scenario. But to implement the contract efficiently, there are participation constraints to satisfy, which are $V_m^{L*} \geq V_m^{D*}$, $\pi_m^{L*} \geq \pi_m^{D*}$, and $\pi_r^{L*} \geq \pi_r^{D*}$. Then we get the proposition 3.

**Proposition 3.** When $0 \leq \alpha < 2 - \sqrt{2}$ and $f_1 \leq F^{L*} \leq f_2$, the socially responsible manufacturer is willing to provide, and the retailer would like to accept the revenue sharing contract with franchise fee.

Where $f_1 = \frac{(1-\theta+c_d-c_r)^2(8+4\alpha+8\alpha\lambda-\alpha^2\lambda-2\alpha^2-16\lambda)}{(2-\alpha)^2(4-\alpha)^2(1-\theta)}$ and $f_2 = \frac{(1-\theta+c_d-c_r)^2(12+8\alpha\lambda-\alpha^2\lambda-4\alpha-16\lambda)}{(2-\alpha)^2(4-\alpha)^2(1-\theta)}$.

To illustrate the optimal franchise fee $F^{L*}$ intuitively, we draw Fig 4 based on $\alpha = 0.2$, $\lambda = 0.2$, $\theta = 0.7$, $c_r = 0.3$, and $c_d = 0.2$.

Proposition 3 implies that when the manufacturer's CSR coefficient is relatively low, revenue sharing contract with franchise fee can coordinate the DCSC. And through franchise fee $F^{L*}$, which is illustrated in Fig 4, the socially responsible manufacturer can make sure supply chain members' profits under low CSR situation are no less than that under D scenario.

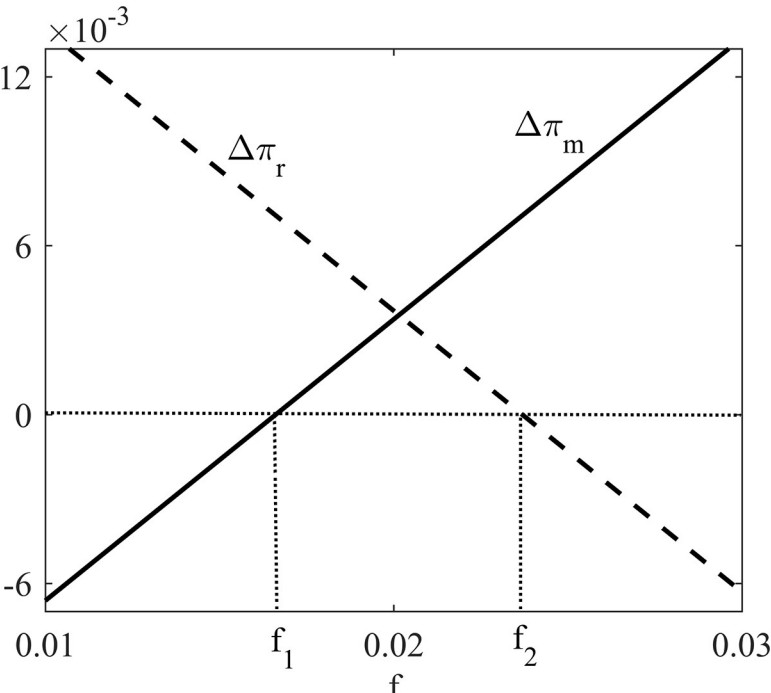

**Fig 4. The optimal franchise fee $F^{L*}$.**

## 5.2. Wholesale price contract with franchise fee and government subsidy under high CSR scenario (H)

When $2 - \sqrt{2} \leq \alpha \leq 1$, internal transfer contract fails to coordinate the DCSC. To obtain the social welfare under C scenario, participation constraints of supply chain members must be satisfied. Besides, the DCSC needs to receive an external subsidy. In China, manufacturers who adopt energy conservation technique and green manufacturing process to fulfill social responsibility, the government always offers tax allowance and exemption or earmarks plenty of money to support them. Based on this, when $2 - \sqrt{2} \leq \alpha \leq 1$, we propose a wholesale price contract with franchise fee and government subsidy for the manufacturer

Under the wholesale price contract with franchise fee and government subsidy, the manufacturer receives a financial subsidy from the government. The subsidy rate is $\varpi$, which means the total subsidy the manufacturer obtained is $\varpi[\alpha - (2 - \sqrt{2})]$. Then the manufacturer charges the retailer a franchise fee $F$ and then commits to lower wholesale price.

Under the wholesale price contract with franchise fee and government subsidy, the optimization problem of the socially responsible manufacturer is:

$$\max_{p_d, w} V_m^H = wq_r + (p_d - c_d)q_d + \alpha CS + \varpi[\alpha - (2 - \sqrt{2})] + F$$
$$\text{s.t. } p_d > w \tag{9}$$
$$\theta p_r > p_d$$

Where,

$$p_r^*(w, p_d) = \arg \max_{p_r} \pi_r^H = (p_r - w - c_r)q_r - F \tag{10}$$

Solving the optimization problem in (9) and (10), we get theorem 4.

**Theorem 4.** Under the wholesale price contract with franchise fee and government subsidy, equilibrium results of the DCSC are: $w^{H*} = \frac{\alpha c_r - \alpha + \theta - c_d}{2 - \alpha}, p_r^{H*} = p_r^{C*}, p_d^{H*} = p_d^{C*}$;

$$\pi_m^{H*} = \frac{(1-\theta)(\theta - c_d)^2 - \alpha[\theta^2(2c_r - 1) + c_d^2] - \alpha\theta[(c_r - 1)^2 - 2c_d c_r]}{\theta(1-\theta)(2-\alpha)^2} + \varpi[\alpha - (2 - \sqrt{2})] + F^{H*},$$

$\pi_r^{H*} = \frac{(1 - \theta - c_r + c_d)^2}{(2 - \alpha)^2(1 - \theta)} - F^{H*}.$

Theorem 4 indicates that when $2 - \sqrt{2} \leq \alpha \leq 1$, the government subsidy can compensate for the loss of the DCSC. It means that wholesale price contract with franchise fee and government subsidy can coordinate the DCSC.

To implement this contract efficiently, there are participation constraints to satisfy, which are $\pi_r^{H*} \geq \pi_r^{D*}, V_m^{H*} \geq V_m^{D*}$ and $\pi_m^{H*} \geq \pi_m^{D*}$. Then we get proposition 4.

**Proposition 4.** When $2 - \sqrt{2} \leq \alpha \leq 1$, $\varpi \geq \varpi_1$, and $F \leq f_3$, the socially responsible manufacturer is willing to provide, and the retailer would like to accept wholesale price contract with franchise fee and government subsidy.

Where $\varpi_1 = \frac{2(1 - \theta - c_r + c_d)^2(4\alpha - \alpha^2 - 2)}{(2-\alpha)^2(1-\theta)(4-\alpha)^2(\alpha - 2 + \sqrt{2})}$ and $f_3 = \frac{4(1 - \theta - c_r + c_d)^2(3 - \alpha)}{(2-\alpha)^2(1-\theta)(4-\alpha)^2}.$

Proposition 4 implies that when CSR coefficient of the manufacturer is relatively high, the subsidy rate of the government is no less than $\varpi_1$ and franchise fee charged by the manufacturer is no bigger than $f_3$, wholesale price contract with franchise fee and government subsidy can achieve coordination. In reality, in response to the petrochemical energy crisis, Chinese government rewards enterprises for their social responsibility in developing new energy resources; to achieve emission reduction and energy saving, the National Development and Reform Commission unites several ministries to implement a project to subsidize the production of energy-efficient products.

# 6. Conclusion

With the emergence of e-commerce, increasing manufacturers with CSR implemented a dual-channel marketing strategy, which breaks the balance between the economic performance and social responsibility performance, and brings uncertainty to the cooperation of the DCSC. For example, the dual-channel strategy brings horizontal competition between traditional and online channels. Will this horizontal competition intensify when the manufacturer has CSR? Besides, CSR usually leads to lower economic performance in the short term, which affects supply chain cooperation. How to mitigate or eliminate it under the dual-channel strategy through a reasonable contract for a socially responsible manufacturer has become a new question in theoretical research and practical management. Considering a DCSC which consists of a socially responsible manufacturer and a retailer, we discuss the influence of CSR coefficient of the manufacturer on equilibrium results and profits under D scenario. Through the comparison of pricing, profits, and social welfare between D and C scenarios, we identify the reason caused supply chain inefficiency under D scenario. Then from the aspect of channel cooperation, we propose two different contracts to achieve coordination of the DCSC.

We find that the manufacturer's CSR is a main factor influencing its pricing strategy. When the manufacturer is more socially responsible, it will cut direct price to promote the increase of consumer surplus and product sales. It suggests the manufacturer that taking consumer surplus into consideration and bearing relatively high CSR can be an effective way to expand market share. We also find that the manufacturer's CSR affects the stability of supply chain structure, which is also the main contribution of this paper. When the manufacturer is less socially responsible, revenue sharing contract with franchise fee can coordinate the DCSC. When the manufacturer has higher socially responsible, the internal transfer contract fails to

achieve coordination, which indicates the dual-channel strategy will be resisted by the traditional retailer. However, due to bearing CSR, the manufacturer may receive an external subsidy. For example, the Chinese government rewards enterprises for their social responsibility in developing new energy resources; Gates Foundation offers subsidy for anti-malarial medicines. In this case, wholesale price contract with franchise fee and government subsidy coordinates the DCSC and obtains the largest social welfare.

## Supporting information

**S1 Data.**
(DOCX)

**S1 Appendix.**
(DOCX)

## Acknowledgments

The authors are grateful to the editors and the three anonymous referees for their constructive and thorough comments, which helps to improve our paper.

## Author Contributions

**Conceptualization:** Shiyang Li.

**Formal analysis:** Shiyang Li.

**Funding acquisition:** Shiyang Li, Nan Zhou.

**Validation:** Nan Zhou.

**Visualization:** Mengli Li.

**Writing – original draft:** Shiyang Li.

**Writing – review & editing:** Mengli Li.

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
