## [Decision Letter · Decision Letter 0]

7 Feb 2020

PONE-D-19-35719

Coordination contract design in a dual-channel supply chain where the manufacturer has corporate social responsibility

PLOS ONE

Dear Dr. Li,

Thank you for submitting your manuscript to PLOS ONE. After careful consideration, we feel that it has merit but does not fully meet PLOS ONE’s publication criteria as it currently stands. Therefore, we invite you to submit a revised version of the manuscript that addresses the points raised during the review process.

We recommend that it should be revised taking into account the changes requested by the reviewers. Since the requested changes includes Major Revision, the revised manuscript will undergo the next round of review by the same reviewers.

We would appreciate receiving your revised manuscript by Mar 23 2020 11:59PM. To enhance the reproducibility of your results, we recommend that if applicable you deposit your laboratory protocols in protocols.io, where a protocol can be assigned its own identifier (DOI) such that it can be cited independently in the future. For instructions see: http://journals.plos.org/plosone/s/submission-guidelines#loc-laboratory-protocols

We look forward to receiving your revised manuscript.

Kind regards,

Baogui Xin, Ph.D.

Academic Editor

PLOS ONE

2. In your data availability statement you write, "N/A." Please ensure you have provided information about the dataset underlying your model and how interested researchers could access it in order to ensure the manuscript meets PLOS' Data Availability policy. If no data underly the modeling presented, please clarify this. Note that the statement will be published if the manuscript is accepted (http://journals.plos.org/plosone/s/data-availability#loc-faqs-for-data-policy).

Reviewers' comments:

Reviewer's Responses to Questions

**Comments to the Author**

1. Is the manuscript technically sound, and do the data support the conclusions?

Reviewer #1: Yes

Reviewer #2: Partly

Reviewer #3: Partly

2. Has the statistical analysis been performed appropriately and rigorously? 

Reviewer #1: Yes

Reviewer #2: Yes

Reviewer #3: N/A

3. Have the authors made all data underlying the findings in their manuscript fully available?

Reviewer #1: Yes

Reviewer #2: Yes

Reviewer #3: No

4. Is the manuscript presented in an intelligible fashion and written in standard English?

Reviewer #1: No

Reviewer #2: Yes

Reviewer #3: No

5. Review Comments to the Author

Reviewer #1: In this paper, in a dual-channel supply chain composed of a socially responsible manufacturer and a retailer, two different contracts are designed to achieve supply chain coordination based on the manufacturer's CSR coefficient, which have certain practical significance. However, the following problems still exist:

1. There are at least two written errors in the manuscript. First, “qh” should be “qr” in equation 5 on page 8. Second, in lines 286 and 287 on page 12, "profit of the DCSC is lower while social welfare is higher than C scenario" should be "profit of the DCSC is lower while social welfare is higher than D scenario ".

2. In the first two parts, the authors propose two coordination contracts in a DCSC in order to maximize social welfare, but the purpose is not clear when the contracts are proposed. The authors need to further clarify the reasons or the purposes of the coordination contracts design. Ideas of the authors should be clearer.

3. As an important variable in this paper, the authors should briefly explain what social welfare means, in order to facilitate the readers' understanding.

4. The analysis of propositions is not deep enough, especially proposition 2. The authors need to further combine the professional knowledge to analyze the causes of the results in order to enhance the persuasiveness of the results.

Reviewer #2: This study considers coordination of a dual-channel SC under the manufacturer’s CSR effort. A wholesale price contract with franchise fee and government subsidy is applied to coordinate the SC.

1- What is the author’s meaning from “too much CSR” in the abstract? How much is “too much”?

2- Some of findings are quit questionable and are not consistent with the literature at all. For example the authors’ model results that: “when the socially responsible manufacturer has high CSR coefficient, the DCSC obtains less revenue under the centralized scenario”. More profit for a centralized supply chain is a rule in the literature of supply chain coordination. Also this is a mathematical fact that the centralized system creates more profit than the decentralized one. I think something is wrong in the developed model which results such an odd outcome.

3- The contribution of this study compared to previous recent studies is quit questionable because some recent publications in this area are missing. Although I know there are some differences between the current work and previous ones, it cannot be guaranteed if the previous literature is not included in your review properly. For example, the developed models should be compared to the following models in the literature review section:

Pricing and greening decisions in a three-tier dual channel supply chain. International Journal of Production Economics, 2019, 217, 185-196.

A collaborative scenario-based decision model for a disrupted dual-channel supply chain. Benchmarking: An International Journal, 2019, doi: 10.1108/BIJ-06-2019-0281

4- It is necessary to show that Hessian matrix (A2) is negative definite by calculating both first and second principal minors.

Reviewer #3: The major defect of the proposed model is that the numerical illustration and sensitivity analysis has not been demonstrated. Also, this study is a lack of managerial implications. The author has used very long sentences due to which it creates a readability problem. Try to make maximum of 11 words.

6. PLOS authors have the option to publish the peer review history of their article (what does this mean?). If published, this will include your full peer review and any attached files.

Reviewer #1: No

Reviewer #2: No

Reviewer #3: No

---

## [Author Response · Author response to Decision Letter 0]

25 Jun 2020

We are grateful to the editors and the three anonymous reviewers for their constructive and thorough comments on our paper. We have seriously considered and addressed all the comments in the revised manuscript. Our point-to-point responses are given in the file named "Response to Reviewers"

---

## [Editor Report · Decision Letter 1]

30 Jun 2020

Pricing and coordination in a dual-channel supply chain with a socially responsible manufacturer

PONE-D-19-35719R1

Dear Dr. Li,

We’re pleased to inform you that your manuscript has been judged scientifically suitable for publication and will be formally accepted for publication once it meets all outstanding technical requirements.

Kind regards,

Baogui Xin, Ph.D.

Academic Editor

PLOS ONE
---

## [Editor Report · Acceptance letter]

6 Jul 2020

PONE-D-19-35719R1 

Pricing and coordination in a dual-channel supply chain with a socially responsible manufacturer 

Dear Dr. Li:

I'm pleased to inform you that your manuscript has been deemed suitable for publication in PLOS ONE. Congratulations! Your manuscript is now with our production department. 

Kind regards, 

on behalf of

Professor Baogui Xin 

Academic Editor

PLOS ONE